# A Multi-Indicator Evaluation Method for Spatial Distribution of Urban Emergency Shelters

**Xinxiang Wang** [1,2,3], **Minglei Guan** [1,4,*] , **Chunlai Dong** [3], **Jingzhe Wang** [1,4], **Yong Fan** [1,4], **Fei Xin** [3] **and Guoyun Lian** [1,4]

1 Institute of Applied Artificial Intelligence of the Guangdong-Hong Kong-Macao Greater Bay Area, Shenzhen Polytechnic, Shenzhen 518055, China
2 Guangzhou Salvage Bureau, Guangzhou 510260, China
3 School of Marine Technology and Geomatics, Jiangsu Ocean University, Lianyungang 222005, China
4 Guangdong Laboratory of Artificial Intelligence and Digital Economy (SZ), Shenzhen 518107, China
* Correspondence: guanminglei@szpt.edu.cn

**Abstract:** Evaluation of the spatial distribution of urban emergency shelters can effectively identify defects in the current distribution of urban emergency shelters and weaknesses in the overall evacuation service capacity of the city and provide reference for improving the level of urban emergency shelters and evacuation and disaster relief capacity. At present, evaluation of the spatial distribution of urban emergency shelters is mainly carried out on three aspects: effectiveness, accessibility, and safety. However, there are problems, such as individual evaluation scales and incomplete indicator systems, unreasonable allocation of indicator weights, and ignoring the influence of fuzzy incompatibility between different indicator attributes on the evaluation results. In this paper, we start from two scales, the individual emergency shelter and the regional groups of emergency shelters. Based on the five criteria of effectiveness, accessibility, safety, suitability, and fairness, the evaluation indicator system of the spatial distribution of urban emergency shelters was constructed. It was combined with AHP, CRITIC, the optimal weight coefficient solution method based on the maximum deviation sum of squares theory, and fuzzy optimization theory to construct a multi-indicator evaluation model. Further, the spatial distribution condition of the existing emergency shelter in Shanghai was evaluated. The results show that: among the existing ninety-one emergency shelters in Shanghai, there are nine places with unreasonable spatial distribution; nineteen places are comparatively unreasonable. From the scale of regional groups, there is one district (Pudong New District) with unreasonable spatial distribution: its relative superiority value is far lower than other districts, and there are three districts that are comparatively unreasonable. Further, the evaluation scores of the spatial distribution reasonableness of emergency shelters in each region of Shanghai show a high–low–middle distribution from the downtown area of Shanghai outward. The evaluation indicator system and evaluation method used in this paper can effectively reflect the deficiencies in the spatial distribution of urban emergency shelters, thus providing a reference for the relevant departments to improve and plan emergency shelters.

**Keywords:** emergency shelter; spatial distribution; multi-indicator evaluation; Shanghai

## 1. Introduction

Urban emergency shelters are used to respond to major natural disasters and emergencies, accepting affected residents for emergency evacuation and helping government organizations to carry out disaster relief work. They are generally established in densely populated and open areas and can take into account the functions of refuge and usual use [1–3]. The construction of urban emergency shelters that are scientifically located and have reasonable layouts can effectively protect the lives of urban residents, reduce casualties, and enhance the city's disaster relief capacity in the event of a disaster [4,5].

Therefore, the evaluation of the spatial distribution of urban emergency shelters can effectively identify the defects of the current urban emergency shelters in terms of location and distribution and the weaknesses of the overall evacuation service capacity of the city, which is of great significance to improve the level of urban emergency shelters and evacuation and disaster relief capacity [6–8].

At present, it is a mainstream evaluation method to analyze and evaluate the spatial distribution of emergency shelters by quantifying the performance of emergency shelters based on multiple indicators and visualizing the results with GIS (geographic information system) tools [9–15]. However, most of the evaluations are based on individual emergency shelters, with a single scale of evaluation, which tends to ignore the spatial complementarity between different shelters and the differences in regional evacuation capacity. As for the selection of evaluation indicators for emergency shelters, it is found that the control indicators for the location and layout of emergency shelters during the construction of emergency shelters are mainly based on the accessibility of residents, safety, and internal service effectiveness [1–3]. On this basis, scholars have established three criteria for accessibility, safety, and effectiveness and expanded the evaluation indicators under the three evaluation criteria. For example, Lu et al. simulated the accessibility of residents through an improved gravity two-step floating catchment area method (G2SFCA) [16]. Wu et al. used the travel distance cost of shelters to measure accessibility [17]. Zhou et al. evaluated the accessibility of urban areas in Beijing and demonstrated that accessibility can effectively evaluate the balance and rationality of the distribution of emergency evacuation shelters [18]. Huang et al. added the minimum distance to the flammable indicator to reflect the safety of the shelter based on the original code [19]. Zhang et al. ensured the safety of the shelter by extracting the slope of the shelter in the process of site selection and planning to avoid areas with geological disasters, such as landslides and mudslides [20]. Xiao indicated the effectiveness of the shelter by the effective evacuation area of the shelter and the effective number of people covered under the service area of the shelter and selected seven shelters for comparative analysis of effectiveness [21]. Xiong evaluated the effectiveness of the place by constructing indicators, such as the practicality of the disaster prevention facilities of the shelter and the accessibility of the internal roads of the shelter [22]. In addition, Liu et al. evaluated the grouping of emergency shelters in different regions based on fairness theory to reflect the service fairness of the shelters [23]. Anhorn et al. proposed an open space ratio indicator to reflect the suitability of the service capacity of emergency shelters [24]. In summary, in the process of multi-indicator evaluation of emergency shelters, previous studies mainly focused on the selection of evaluation indicators regarding the three evaluation aspects of accessibility, safety, and effectiveness, and the selection of evaluation objects was mostly based on the evaluation of single emergency shelters, ignoring the impact of the cooperation and functional complementarity between different emergency shelters on the overall regional sheltering capacity. Although Liu and Anhorn et al. proposed and proved the importance of fairness and suitability in the evaluation of emergency shelters, they did not add them to the multi-indicator evaluation system for systematic evaluation. To solve the problem, this paper divides the evaluation of the spatial distribution of urban emergency shelters into two scales: individual shelters and regional groups of shelters. In the establishment of the evaluation indicator system, the indicators under the five aspects of effectiveness, accessibility, safety, suitability, and fairness of the emergency shelter are selected from the elements related to the spatial distribution of the emergency shelter to quantitatively reflect the advantages and disadvantages of the emergency shelter in terms of spatial distribution. In terms of the evaluation methods for emergency shelters, the commonly used methods include analytic hierarchy process (AHP) [25–27], grey correlation analysis [28–30], technique for order preference by similarity to an ideal solution (TOPSIS) [31], and real coded accelerated genetic algorithm——projection pursuit (RAGA-PP) [32,33]. The AHP method can divide and rank the importance of multiple indicators in a hierarchy, but it relies too much on the subjective experience of decision-makers, which may result in weighting deviations from the actual situation and affect

the final evaluation results. The grey correlation analysis method and RAGA-PP method mainly rely on the numerical correlation and numerical structure of data samples to rank the importance of each indicator. Once the data are wrong or missing, it is easy to affect the final evaluation result. The above methods do not take into account the influence of fuzzy incompatibility between different attribute indicators on the evaluation results in multi-indicator decision-making. To solve the problem, this paper combines the established evaluation indicator system with multi-indicator decision-making and fuzzy optimization based on fuzzy theory as a way to overcome the influence of fuzziness and uncertainty between different indicator attributes and establish a quantitative evaluation model of urban emergency shelters. In the optimization of indicator weight assignment of the model, this paper determines the subjective and objective weights by AHP and CRITIC (criteria importance though intercriteria correlation) and solves the optimal weight coefficients of the subjective and objective combination assignment by the theory of maximum deviation sum of squares to determine the optimal weights of the evaluation indicators and make the model's comprehensive evaluation of the spatial distribution condition of urban emergency shelters more scientific and reasonable.

This research proposes a new multi-indicator evaluating framework for the spatial distribution of emergency shelters. The framework constructs an evaluation indicator system based on five criteria aspects of effectiveness, accessibility, safety, suitability, and fairness, quantifies evaluation indicators through big data acquired by open resources, optimizes indicator weights by the AHP method and CRITIC method, and establishes a quantitative evaluation model for the spatial reasonableness of emergency shelter distribution by combining the fuzzy optimization method to provide support for the planning and optimization of urban emergency shelters. In this paper, the applicability of the method is analyzed by taking the Shanghai emergency shelter as the research object. According to the results of the research, planners can identify the weaknesses of the urban existing disaster prevention and evacuation resources and plan and improve them to address the deficiencies in the urban current emergency evacuation service capacity so as to maximize the efficiency of the urban existing disaster prevention and evacuation resources, enhance urban safety, and promote social harmony and fairness.

## 2. Material and Methods

This paper proposes a multi-indicator evaluation framework for the spatial distribution of urban emergency shelters through urban big data combined with GIS technology, which mainly includes four parts: dataset construction of the study area, quantification of evaluation indicators, indicator weight calculation, and evaluation score calculation, as shown in Figure 1.

### 2.1. Study Area

This paper takes Shanghai as the study case area. Shanghai is located between $120°52'\sim122°12'$E longitude and $30°40'\sim31°53'$N latitude and is a coastal city in a low-altitude area. It is also China's international economic, financial, trade, shipping, and technology innovation center, and is ranked third in the world in the Global Financial Center Index (GFCI). As one of the seven mega-cities in China, Shanghai has a resident population of nearly 25 million people. With its high population density, building density, and resource intensity, the range of damage caused by natural disasters and the economic and population losses incurred are also larger. Potential natural disasters in Shanghai include typhoons, heavy rainfall and flooding, lightning strikes, ground subsidence, earthquakes, etc. [34]. As emergency evacuation sites established in response to major natural disasters and emergencies, urban emergency shelters can effectively protect people's lives, reduce casualties, enhance the city's evacuation and disaster relief capabilities, and provide assistance for post-disaster recovery. According to the information released by the Shanghai Municipal Bureau of Statistics, as of 2020, Shanghai has built a total of 91 emergency shelters available for use, including 2 Class I emergency shelters, 60 Class II emergency

shelters, and 29 Class III emergency shelters. The regional overview in Shanghai is shown in Figure 2. The control of indicators of effective shelter area per capita and service radius of different levels of emergency shelters are shown in Table 1.

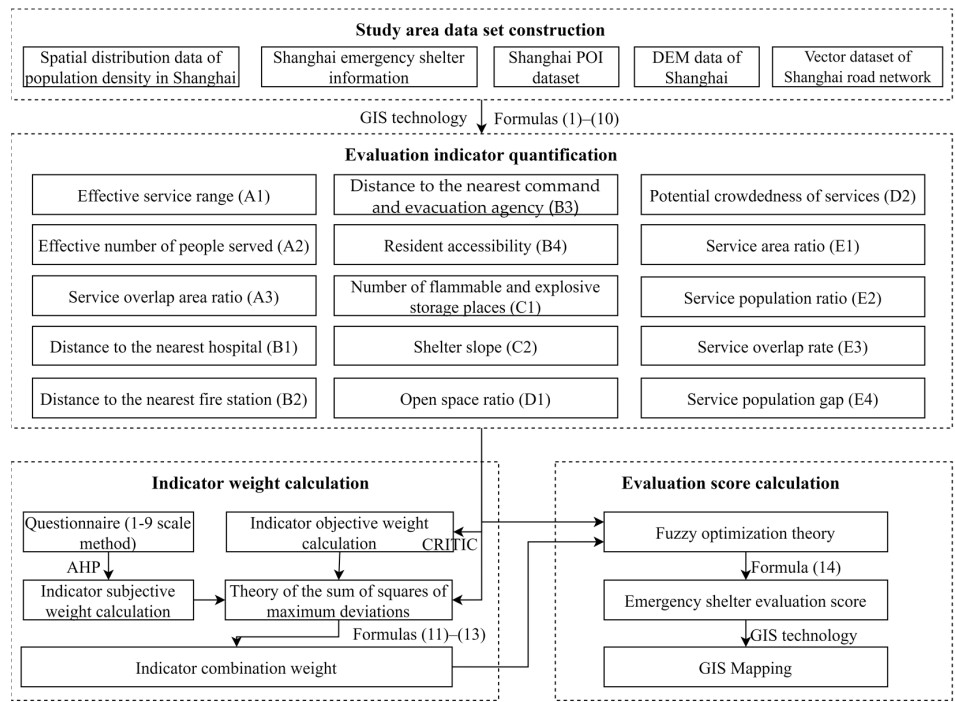

**Figure 1.** Technology roadmap.

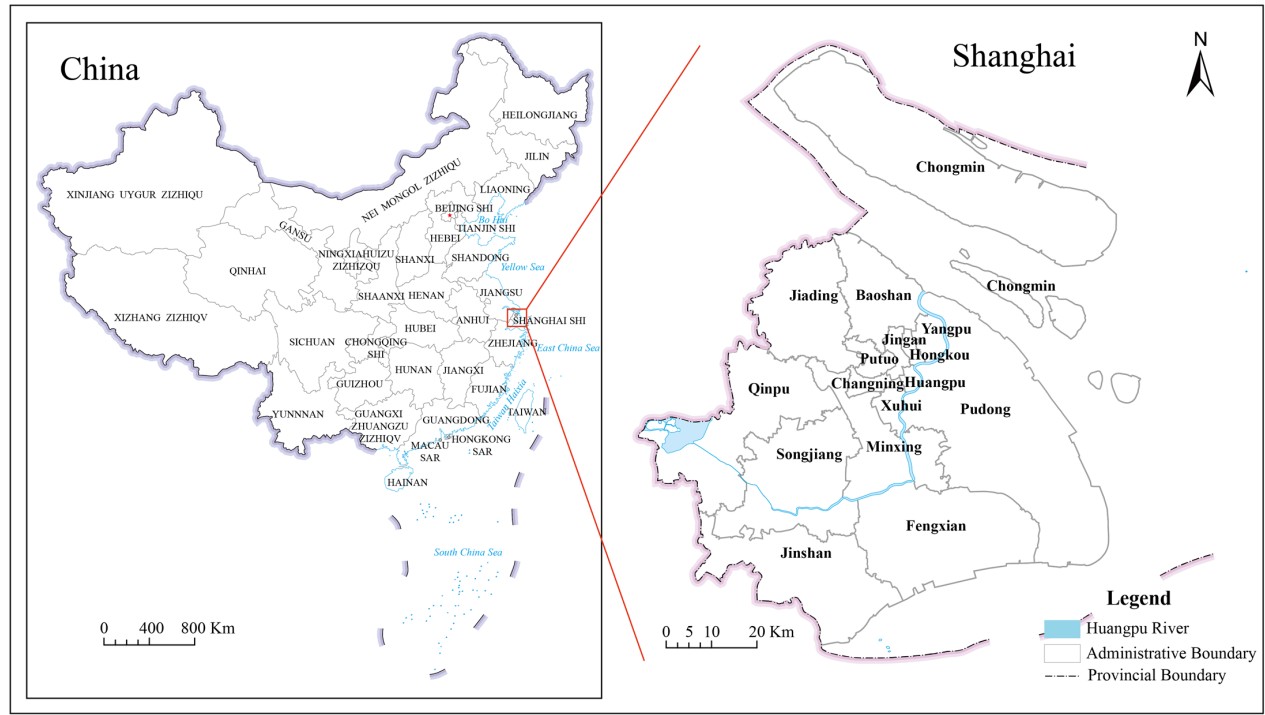

**Figure 2.** Overview of the study area.

**Table 1.** Hierarchical control requirements for emergency shelters in Shanghai.

| Emergency Shelter Classification | Types | Effective Shelter Area per Person (m$^2$) | Service Radius (m) |
|---|---|---|---|
| Class I | Space Construction | ≥3.0<br>2.0~3.5 | 5000 |
| Class II | Space Construction | ≥2.0<br>1.5~2.0 | 1000 |
| Class III | Space Construction | ≥1.5<br>1.0~1.5 | 500 |

The table data are from "Design Standard for Emergency Shelter (DG/TJ 08-2188-2015)".

### 2.2. Study Datasets

The data sources used in this study are shown in Table 2. Among them, four data items, namely Shanghai Emergency Shelter Basic Information, Shanghai Secondary Medical Institutions, Map of China's Provinces-Shanghai, and Shanghai Average Population per Household, are officially released data, while the rest are all open resource data. The official data are highly reliable, but the available data are limited, which is difficult to meet the needs of research. Open resource data has a wide range of data sources and easy data access, but its data reliability needs to be verified. For example, in this paper, we use the vector data of the road network provided by OpenStreetMap, an open-source map service platform, but the results based on this vector data need to be overlaid on the satellite remote sensing image map to verify. For the point data obtained from the open map platform (Amap), the advantage is that it improves the accessibility of data and has high data reliability, but this type of data has single attribute information and only has basic location information, and, since the data is obtained based on keywords and data classification codes, data selection and sampling are needed to reduce the redundancy and improve the accuracy of data. For population distribution data, previous studies have used neighborhoods or streets as demographic units when conducting population distribution simulations [17,35]. The scale of the statistical unit of this method is too large and the population is concentrated in the center of mass of the community or street, and the spatial distribution of the population is not balanced. In this paper, the population distribution data of Shanghai take the residential area as the demographic unit, which expresses the total population of each residential area by the product of the number of households and the population of each household and obtains the density analysis (Figure 3). This method has a moderate scale of demographic units, a more reasonable population distribution, and is suitable for a large study area. The shelter slope of the Shanghai emergency shelter was extracted from ASTER GDEM 30M resolution DEM data using the slope analysis function of ArcGIS.

**Table 2.** Data list.

| Target Data | Original Data | Data Format | Access Source |
|---|---|---|---|
| 2019 Shanghai Emergency Shelter Distribution Data | Basic Information on Emergency Shelters in Shanghai (First Quarter of 2020) | Point Data | the official website of Shanghai Municipal Statistics Bureau (http://tjj.sh.gov.cn/, accessed on 17 October 2020) |
| Shanghai Population Distribution Data | Shanghai residential area location data | Point Data | Amap API developer platform (https://lbs.amap.com/, accessed on 29 April 2021) |
|  | Average population per household in Shanghai | Attribute Data | Shanghai Statistical Yearbook 2020 (http://tjj.sh.gov.cn/, accessed on 7 February 2022)) |
|  | Number of households in Shanghai residential area | Attribute Data | lianjia, Shell, and other housing websites (https://gz.lianjia.com/, accessed on 6 February 2022; https://gz.ke.com/, accessed on 6 February 2022) |

**Table 2.** *Cont.*

| Target Data | Original Data | Data Format | Access Source |
|---|---|---|---|
| Shanghai Command and Evacuation Agency Distribution Data | Location data of public security bureau, local police station, traffic police | Point Data | Amap API developer platform (https://lbs.amap.com/, accessed on 16 April 2021) |
| Shanghai Fire Station Distribution Data | Location data of fire stations | Point Data | Amap API developer platform (https://lbs.amap.com/, accessed on 5 April 2021) |
| Shanghai Flammable and Explosive Storage Place Distribution data | Location data of gas stations and warehouses | Point Data | Amap API developer platform (https://lbs.amap.com/, accessed on 15 April 2021) |
| Distribution of hospitals above the second level in Shanghai | Shanghai Secondary Medical Institutions | Point Data | the official website of the Shanghai Municipal Health Commission (http://wsjkw.sh.gov.cn/, accessed on 16 April 2021) |
| Shanghai Road Network Vector Data | china-latest-free.shp | Line Data | OpenStreetMap (https://www.openstreetmap.org/, accessed on 27 February 2022) |
| Shanghai Administrative Map | Map of China's Provinces-Shanghai (Review No. GS(2019)3333) | Polygon Data | the standard map service system (http://bzdt.ch.mnr.gov.cn/, accessed on 26 July 2021) |
| Shanghai Slope Data | ASTER GDEM 30M resolution DEM data | Polygon Data | Geospatial Data Cloud (https://www.gscloud.cn/, accessed on 11 May 2021) |

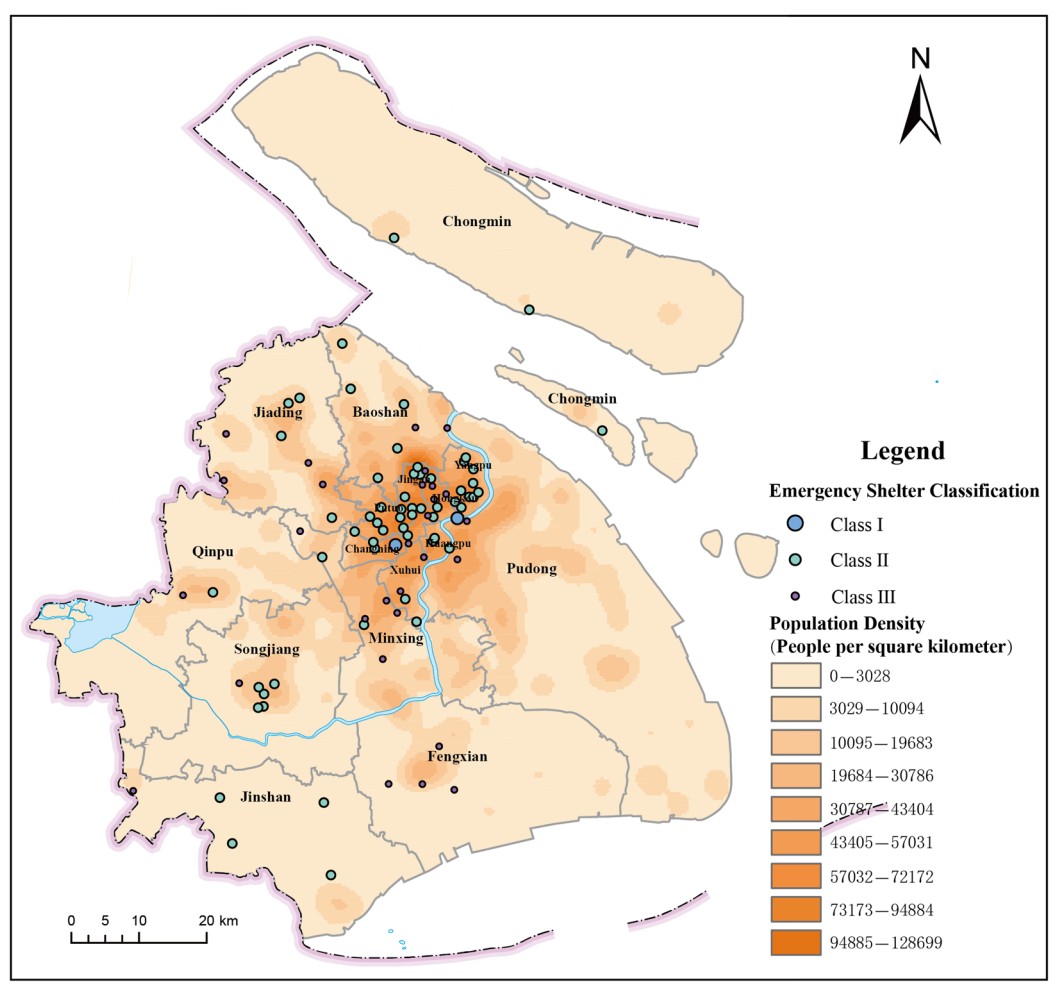

**Figure 3.** Spatial distribution of the population density in Shanghai.

*2.3. Evaluation Indicators Selection and Quantification*

This paper analyzes and evaluates the reasonableness of the spatial distribution of emergency shelters from two scales: single emergency shelters and regional groups of emergency shelters. Compared with previous studies, this paper summarizes and expands the evaluation criteria under the evaluation scale of a single emergency shelter. The evaluation indicator under the three criteria of effectiveness, accessibility, and safety was improved, and the suitability criterion was added to reflect the suitability of emergency shelters for evacuation. At the same time, to reflect the strength and problems of the emergency shelter service capacity in different regions, this paper also starts from the evaluation scale of the regional groups of emergency shelters. By grouping the emergency shelters within the region, the advantages and disadvantages of the refuge service capacity between different regions can be reflected through the fairness criteria to reflect the reasonableness of the spatial distribution of the emergency shelters in each region.

(1)  Effectiveness

Traditional effectiveness criteria are evaluated by the effective shelter area inside the emergency shelter. This paper evaluates the spatial distribution of shelters, so the evaluation indicators are constructed by combining the service radius of shelters and the accessibility of the road network. The actual service supply capacity of emergency shelters under the current spatial distribution is reflected by three indicators: the effective service range, the effective number of people served, and the service overlap area ratio.

The effective service range refers to the total area of the service range of the emergency shelter combined with the accessibility of the road network and the service radius of the shelter, which is used to reflect the size of the actual service range of the shelter.

The effective number of people served refers to the total number of evacuees covered under the effective service range of emergency shelters, which is used to reflect the number of people covered under the actual range of services of the shelter.

The service overlap area ratio is used to reflect the redundancy of emergency shelter spatial services, and its quantitative formula is shown in Equation (1).

$$SOR_j = \frac{SO_j}{SA_j} \times 100\%, \tag{1}$$

where: $SOR_j$ indicates the service overlap area ratio of emergency shelter $j$; $SO_j$ indicates the service overlap area of emergency shelter $j$ and other places; $SA_j$ indicates the total service area of emergency shelter $j$.

(2)  Accessibility

The accessibility criterion is used to reflect the timeliness of disaster relief and the accessibility of urban emergency shelters under the current spatial distribution. Under the accessibility criterion of the traditional evaluation method, four accessibility indicators are generally included: the distance between the emergency shelter and the nearest hospital, fire station, command and evacuation agency, and residential area. In the measurement of the accessibility of residents, since residents are the main body of urban emergency shelter services, it is unreasonable to measure only by distance and should take into account the imbalance between the number of residents in need of shelter and the number of emergency shelter services supplied.

Two-step floating catchment area (2SFCA) is a commonly used method to measure accessibility. The traditional two-step mobile search method calculates accessibility values centered on the residents' area (demand points) [36,37]. This paper discusses the accessibility of emergency shelters, so the comprehensive accessibility value for residents of each emergency shelter is calculated by taking the emergency shelter (supply point) as the center and the service radius of the emergency shelter as the search range, and its

calculation formula is shown in Equation (2). The higher the accessibility score, the better the accessibility of the emergency shelter to the residents.

$$A_j = \sum_{i=1}^{n} A_{ij} = \sum_{i=1}^{n} \left[ \frac{D_j g(d_{ij})}{\sum_{i=1}^{n} D_i g(d_{ij})} \right], \qquad (2)$$

where: $A_j$ denotes the comprehensive accessibility value of residents of emergency shelter $j$; $A_{ij}$ denotes the accessibility between places $i$ and $j$; $D_j$ denotes the effective number of people accommodated in place $j$; $D_i$ denotes the population size of residential area $i$; $n$ denotes the number of residential areas; and $g(d_{ij})$ is the distance decay function between the two places.

When a disaster occurs, the probability of people arriving at the emergency shelter is inversely proportional to its distance, and, the greater the distance, the stronger the attenuation effect, which is in line with the curve change trend of the kernel density function. Therefore, this paper uses the kernel density function as the decay function, and its calculation formula is shown in Equation (3).

$$g(d_{ij}) = \frac{3}{4} \left[ 1 - \left( \frac{d_{ij}}{d_0} \right)^2 \right], \quad (d_{ij} \leq d_0) \qquad (3)$$

where: $d_{ij}$ denotes the shortest network distance between two points; $d_0$ is the distance threshold; and here denotes the service radius of the emergency shelter $j$.

(3)　Safety

Safety criteria consider the disasters that secondary disasters or potential disasters may cause to emergency shelters. It is stipulated in the code that the emergency shelter should be far from dangerous areas and the distance between the flammable and explosive places should be more than 1000 m, and the slope of the shelter should not be more than 10%. Therefore, this paper takes the number of flammable and explosive storage places within a kilometer of the emergency shelter and the slope of the shelter as the evaluation indicator of spatial distribution safety criteria. The more flammable and explosive storage places within a kilometer of the shelter, the greater the slope of the shelter, the lower the safety of the shelter, and the more unreasonable the space distribution.

(4)　Suitability

Open space ratio is a common indicator to evaluate the suitability of the shelter, and its quantification formula is the ratio of the effective shelter area to the open area of the shelter. The higher the ratio of open space ratio, the higher the potential service efficiency of the shelter.

The potential crowdedness of services is an important indicator for evaluating public places and can be used to reflect the resource allocation gap of public service facilities [38,39]. However, the traditional suitability criterion is not included, so this paper adds the service potential crowdedness indicator to the suitability criterion to enrich the evaluation dimension of the indicator. Inverted two-step floating catchment area (i2SFCA) is a commonly used method to estimate the potential crowdedness of facilities [40,41]. In this paper, the method is improved by referring to the multiplicative competitive interaction model (MCI). The number of command and evacuation agencies is added as an attribute of the competitiveness of the shelter, and the service radius and relative accessibility of the emergency shelter are combined to reflect the choice of evacuation behavior and the accessibility of the residents. The formula for calculating the potential crowdedness of emergency shelters is shown in Equation (4).

$$C_j = \frac{\sum_{i=1}^{n} F_{ij}}{S_j} = \frac{\sum_{i=1}^{m} D_i P_{ij} R_{ij}}{S_j}, \quad (i \in \{d_{ij} \leq d_0\}) \qquad (4)$$

where: $C_j$ denotes the potential crowdedness of emergency shelter $j$, and its unit is people/m$^2$; $F_{ij}$ denotes the potential number of people served by shelter $j$ to residential area $i$; $S_j$ is the effective shelter area of emergency shelter $j$; $P_{ij}$ denotes the attractiveness of emergency shelter $j$ to residential area $i$, and the calculation formula is shown in Equation (5); $R_{ij}$ denotes the relative accessibility between residential area $i$ and shelter $j$, and the calculation formula is shown in Equation (6).

$$P_{ij} = \frac{S_j(E_j + 1)g(d_{ij})}{\sum_{k=1}^{m} S_k(E_k + 1)g(d_{ik})} , \tag{5}$$

where: $P_{ij}$ denotes the attractiveness of emergency shelter $j$ to the residents of area $i$; $E_j$ denotes the number of command evacuation agencies within the service area of emergency shelter $j$; $m$ denotes the number of emergency shelters in the study area.

$$R_{ij} = \begin{cases} 1 & , \left(\frac{A_{ij}}{\overline{A}} \geq 1\right) \\ A_{ij}/\overline{A} & , \left(\frac{A_{ij}}{\overline{A}} < 1\right) \end{cases} \tag{6}$$

where: $R_{ij}$ denotes the relative accessibility between place $i$ and place $j$; $\overline{A}$ enotes the average accessibility of all residential areas under the service area of emergency shelter $j$.

(5)  Fairness

The construction and service capacity of emergency shelters in different regions of the city are limited by the level of regional economic development, and there is a certain gap. For the evacuation residents, the reasonableness of the spatial distribution of the regional emergency shelter grouping also means fairness when evacuating. Regional emergency shelter grouping refers to the combination of emergency shelters in the same region to eliminate overlapping services of different shelters so as to truly reflect the actual sheltering capacity of different regions. This paper constructs fairness criteria through four indicators: service area ratio, service population ratio, service overlap rate, and service population gap to reflect the reasonableness of resource supply capacity in spatial allocation after grouping of emergency shelters in different regions.

The service area ratio is used to reflect the spatial service capacity of emergency shelters after regional grouping. Its calculation formula is shown in Equation (7).

$$SAR_k = \frac{\sum_{j \in k} \cup SA_j}{A_k} \tag{7}$$

where: $SAR_k$ denotes the service area ratio after the grouping of emergency shelters in region $k$; $SA_j$ denotes the area of each emergency shelter service range; $A_k$ denotes the area of region $k$.

The service population ratio is used to reflect the service capacity of the regional evacuation population after the regional grouping of emergency shelters. Its calculation formula is shown in Equation (8).

$$SPR_k = \frac{\sum_{j \in k} \cup SD_j}{P_k} \tag{8}$$

where: $SPR_k$ denotes the ratio of population served after the grouping of emergency shelters in region $k$; $SD_j$ denotes the number of population covered under the service area of each emergency shelter; $P_k$ denotes the total population of region $k$.

Service overlap rate reflects the redundancy of spatial services of regional shelters and the mutual reinforcement of shelters between different regions. Its calculation formula is shown in Equation (9).

$$SOR_k = \frac{\sum_{j \in k} SA_j - \sum_{j \in k} \cup SA_j}{\sum_{j \in k} \cup SA_j} \tag{9}$$

where: $SOR_k$ denotes the service overlap rate after the grouping of emergency shelters in region $k$; $\sum_{j \in k} SA_j - \sum_{j \in k} \cup SA_j$ denotes the service overlap area after the grouping of shelters in region $k$.

The service population gap reflects the size of the current emergency shelter service gap in the region. Its calculation formula is shown in Equation (10).

$$SPG_k = P_k - \sum_{j \in k} \cup SD_j \tag{10}$$

where: $SPG_k$ denotes the population gap served by the grouped emergency shelters in region $k$; $P_k$ denotes the total population of region $k$; $\sum_{j \in k} \cup SD_j$ denotes the total number of people that can be served by the grouped emergency shelters in region $k$.

The established multiple evaluation indicator system for the reasonableness of the spatial distribution of urban emergency shelters is shown in Table 1. The quantification of evaluation indicators is shown in Equations (1) to (10), and the variables in the equations are obtained by combining the raw data obtained in Table 3 through the spatial analysis function of ArcGIS. As an example, Equation (1) involves two variables: the service area of the shelter and the service overlap area. The service radius of different classes of shelters is different; for example, the service radius of class I of shelters is 5000 m. The service area analysis function under the network analysis of ArcGIS can calculate the service range and service area of class I of emergency shelter based on the current road network and the service radius of 5000 m. Similarly, the service area of different classes of shelters can be obtained by the same operation. For emergency shelter j, calculate the area of the intersection of its service area with other shelters (the intersection function of ArcGIS) to obtain the service overlap area of the shelter. The same calculation for other shelters.

**Table 3.** Evaluation indicator system.

| Target Layer | Criterion Layer | Indicator Layer |
|---|---|---|
| The Spatial Distribution Reasonableness of Individual Emergency Shelters (T1) | Effectiveness (A) | Effective service range (A1) Effective number of people served (A2) Service overlap area ratio (A3) |
| | Accessibility (B) | Distance to the nearest hospital (B1) Distance to the nearest fire station (B2) Distance to the nearest command and evacuation agency (B3) Resident accessibility (B4) |
| | Safety (C) | Number of flammable and explosive storage places (C1) Shelter slope (C2) |
| | Suitability (D) | Open space ratio (D1) Potential crowdedness of services (D2) |
| The Spatial Distribution Reasonableness of Regional Groups of Emergency Shelters (T2) | Fairness (E) | Service area ratio (E1) Service population ratio (E2) Service overlap rate (E3) Service population gap (E4) |

## 2.4. Indicator Weights Model Construction

The reasonableness of indicator weights determines the accuracy of the spatial distribution of emergency shelters in the best order. In this paper, the AHP and CRITIC methods are used to calculate the subjective and objective weights of each indicator according to the characteristics of evaluation indicators. Further, the optimal combination weight of the evaluation indicator is obtained by solving the optimal weight coefficient of the combination assignment through the maximum deviation sum of squares theory.

### 2.4.1. Indicator Subjective Weight Calculation

The evaluation of the spatial distribution of urban emergency shelters is a multi-attribute decision problem, and the hierarchical structure characteristics of its evaluation

indicator system are shown in Table 3. The analytic hierarchy process (AHP) method can decompose target attributes in a hierarchical structure and rank the importance of different elements belonging to the same layer by comparing them with each other. Further, it is suitable for decision-making on the evaluation of the spatial distribution of urban emergency shelters. The calculation steps can be divided into four steps: the first step is to establish the recursive hierarchy model (Table 3); the second step is to construct the judgment matrix according to the 1–9 scale method; the third step is to calculate the single-level weights and consistency test for the constructed judgment matrix; the fourth step is to calculate the subjective weights of each indicator [25,26].

### 2.4.2. Indicator Objective Weight Calculation

The CRITIC method is an objective weighting method that is more commonly used and reflects the information of the data itself more than other objective weighting methods. Its calculation steps are generally divided into the following four steps:

The first step is to establish the original data matrix. The second step is to assimilate the indicator data, and this paper uses the forwarding process to convert all the indicator types to maximize indicators. The third step is to calculate the indicator variability and conflict. The fourth step is to determine the objective weight according to the amount of information contained in the indicator [42–44].

### 2.4.3. Determination of Optimal Weight Coefficients for Combination Weighting

The computational steps of the optimal weight coefficient solution method based on the maximum deviation sum of squares theory for the combination weighting can be divided into the following four steps [45–47].

In the first step, the relative affiliation matrix Y is established by forwarding and inverting the raw data of each evaluation indicator according to the characteristics that the larger is better or the smaller is better.

The second step is to construct the combined weighting coefficient vector and the weight block matrix.

$$W = \left(w_j\right)_{1 \times n} = W_{A-C} L = (W_{AHP}, W_{CRITIC})(l_1, l_2)^T \tag{11}$$

where: $W$ denotes the combination weight matrix; $w_j$ denotes the combination weight of the indicator $j$; $W_{A-C}$ denotes the subjective and objective weight blocking matrix; $L$ denotes the subjective and objective weight coefficient matrix and satisfies $\sum_{i=1}^{2} l_i^2 = 1$.

The third step is to construct the objective decision function based on the principle of maximizing the sum of squares of the total deviations of the objective.

The fourth step is to solve the weight coefficient of combination weighting. According to the objective decision function, the deviation square sum optimal combination weighting model (Equation (12)) is established. The eigenvector $L' = \left(l'_k\right)_{1 \times k}^T$ is obtained by solving the optimal solution of the model, and the optimal weight coefficient is obtained by normalization of $L'$.

$$maxF(L) = \frac{L^T W_{A-C}^T Y' W_{A-C} L}{L^T L} , \tag{12}$$

$$Y' = \left(y'_{ij}\right)_{n \times n} = \left(\sum_{k_1=1}^{m} \sum_{k_2=1}^{m} \left(y_{k_1 i} - y_{k_2 i}\right)\left(y_{k_1 j} - y_{k_2 j}\right)\right)_{n \times n} \tag{13}$$

where: $Y'$ is a symmetric non-negative definite sum of squares of $n \times n$ order; $y'_{ij}$ denotes the sum of squares of deviations between objective $i$ and objective $j$, $i, j \in \{1, 2, \ldots, n\}$.

### 2.5. Multi-Indicator Evaluation Model Construction

The evaluation of the reasonableness of the spatial distribution of emergency shelters is a multi-indicator decision problem, but the value range of different attribute indicators is fuzzy and uncertain in the evaluation. Fuzzy theory can solve this problem well and obtain the relative optimal solution of the decision problem [48–50]. The steps of constructing a

multi-indicator decision evaluation model based on fuzzy theory can be roughly divided into the following three steps.

The first step is to establish the relative superiority matrix. The second step is to establish the relative superiority function model. In the third step, the relative superiority function model is derived to solve the optimal solution to obtain the multi-indicator decision evaluation score $\mu_i$. The higher the value of $\mu_i$, the more reasonable the spatial distribution of the evaluation object.

$$\mu_i = \frac{1}{1 + \left[ \frac{\sum_{j=1}^{n} \left( w_j \left| y_{ij} - 1 \right| \right)^p}{\sum_{j=1}^{n} \left( w_j \left| y_{ij} \right| \right)^p} \right]^{\frac{2}{p}}} \tag{14}$$

where: $\mu_i$ denotes the relative superiority of evaluation object $i$; $i \in \{1, 2, \ldots, m\}$; $j \in \{1, 2, \ldots, n\}$; $p$ is the distance parameter, generally taken as $p = 2$; $\mu_i$ is the value of the relative superiority of the evaluation object $j$.

## 3. Results and Analysis

### 3.1. Quantitative Results of Indicators

According to the definitions of evaluation indicators in this paper, the evaluation indicators of emergency shelters are quantified by combination with the spatial analysis function of ArcGIS, and the quantified values of evaluation indicators under the four criteria of effectiveness, accessibility, safety, and suitability are shown in Table 4.

**Table 4.** Evaluation indicator data of some emergency shelters.

| Emergency Shelter Serial Number | A1 (km²) | A2 (10⁴) | A3 (%) | B1 (km) | B2 (km) | B3 (km) | B4 | C1 | C2 (%) | D1 (%) | D2 |
|---|---|---|---|---|---|---|---|---|---|---|---|
| 1 | 1.735 | 14.41 | 0 | 1.143 | 0.883 | 0.082 | 1.47 | 2 | 2 | 60.94 | 75.48 |
| 2 | 1.195 | 2.31 | 0 | 2.729 | 25.12 | 0.845 | 15.15 | 0 | 6 | 64.47 | 15.22 |
| 3 | 1.671 | 17.11 | 90.88 | 0.521 | 1.821 | 0.419 | 1.70 | 2 | 3 | 37.04 | 6.98 |
| 89 | 0.348 | 2.15 | 0 | 1.963 | 4.155 | 1.124 | 2.54 | 0 | 5 | 66.67 | 0.03 |
| 90 | 1.516 | 5.84 | 0 | 8.668 | 4.452 | 1.774 | 1.45 | 1 | 3 | 30.18 | 0.78 |
| 91 | 1.559 | 15.88 | 100 | 1.502 | 1.082 | 1.211 | 1.57 | 2 | 5 | 73.33 | 0.66 |

According to the effectiveness quantification results, nearly 57 shelters have an effective number of people covered under the effective range of services between 10,000 and 60,000, with 22 shelters exceeding this value, and the overall service capacity of shelters in the region is good. However, there are 12 places in which the service overlap area ratio is close to 100%, and the spatial service redundancy is high, which needs to be improved.

According to the accessibility quantification results, the accessibility between emergency shelters and the three service providers in the downtown of Shanghai is more advantageous than in other areas, but there are many places with poor comprehensive accessibility values of residents.

According to the safety quantification results, there are seventy shelters within a kilometer of the existence of flammable and explosive places, there are three emergency shelters within one kilometer of its flammable and explosive sites number more than 10, and there are two shelters with a slope higher than 10% that do not meet the requirements of the code. The security situation at most shelters in the study area is not good.

According to the suitability quantification results, only five shelters have an open space ratio lower than 10%, and the degree of open space of places in the region is good. However, the potential crowdedness of twenty-nine emergency shelters exceeds three persons per square meter, and most of the emergency shelters with high crowdedness are concentrated in the downtown area of Shanghai.

The emergency shelters in the study area are divided according to administrative districts, the shelters in the same district are grouped, and the quantitative values of each

evaluation indicator under the fairness criterion of emergency shelters were calculated based on the spatial analysis function of ArcGIS, as shown in Table 5.

**Table 5.** Fairness indicators of emergency shelters in each district of Shanghai.

| District | Number of Emergency Shelters | E1 (%) | E2 (%) | E3 (%) | E4 ($10^4$) |
|---|---|---|---|---|---|
| Huangpu | 4 | 33.18 | 18.82 | 14.37 | 52.83 |
| Xuhui | 4 | 19.54 | 27.15 | 0.00 | 79.74 |
| Changning | 2 | 44.92 | 97.85 | 3.65 | 14.92 |
| Jingan | 11 | 46.13 | 98.02 | 15.60 | 2.10 |
| Putuo | 10 | 40.64 | 78.29 | 17.56 | 27.70 |
| Hongkou | 5 | 60.28 | 66.45 | 12.58 | 26.64 |
| Yangpu | 12 | 46.81 | 78.71 | 20.63 | 27.78 |
| Minxing | 6 | 1.47 | 5.05 | 0.00 | 242.04 |
| Baoshan | 8 | 2.40 | 8.49 | 0.71 | 187.07 |
| Jiading | 8 | 1.73 | 8.62 | 0.00 | 145.85 |
| Pudong | 1 | 0.57 | 4.94 | 0.00 | 529.21 |
| Jinshan | 5 | 0.58 | 2.94 | 0.00 | 78.33 |
| Songjiang | 6 | 1.14 | 4.98 | 1.60 | 168.37 |
| Qingpu | 2 | 0.29 | 2.02 | 0.00 | 120.81 |
| Fengxian | 4 | 0.18 | 3.12 | 0.00 | 112.16 |
| Chongming | 3 | 0.25 | 3.28 | 0.00 | 66.12 |

The fairness quantification results show that Hongkou District has the highest service area ratio (60.28%). Fengxian District was the lowest, with only 0.18%. The service overlap rate of Xuhui, Minxing, Jiading, Pudong, Jinshan, Qingpu, Fengxian, and Chongming districts is 0%, indicating high spatial service effectiveness. The ratios of serving population in Changning and Jing'an districts are over 90%; the population service capacity is relatively greater, followed by Putuo, Hongkou, and Yangpu districts, which are between 60% and 80%, respectively. The largest service gap is currently in Pudong, which accounts for 28% of the city's total service gap.

It can be seen from the number of service population gaps that the number of emergency shelters in each district of Shanghai still cannot meet the demand of the evacuation population, except for Changning, Jing'an, Putuo, Hongkou, and Yangpu, which have high population service capacity. The rest of the districts have a large service population gap, and there is an urgent need to plan new emergency shelters to improve the overall and regional evacuation service capacity of Shanghai.

### 3.2. Indicator Weighting

A single-level judgment matrix was constructed according to Table 1, and the subjective weights of each indicator were calculated by combination with the AHP method. The objective weights of each evaluation indicator were calculated by the CRITIC method. Finally, the relative affiliation matrix was established according to the type of indicator, and the deviation square sum optimal combination weighting model was established. By solving the model, the optimal weight coefficient matrix $L_{T1} = [0.4398, 0.5602]^T$ and $L_{T2} = [0.4794, 0.5206]^T$ for the subjective–objective combination assignment of the target layers T1 and T2. The subjective weight, objective weight, and combined weight of each evaluation indicator are shown in Figure 4.

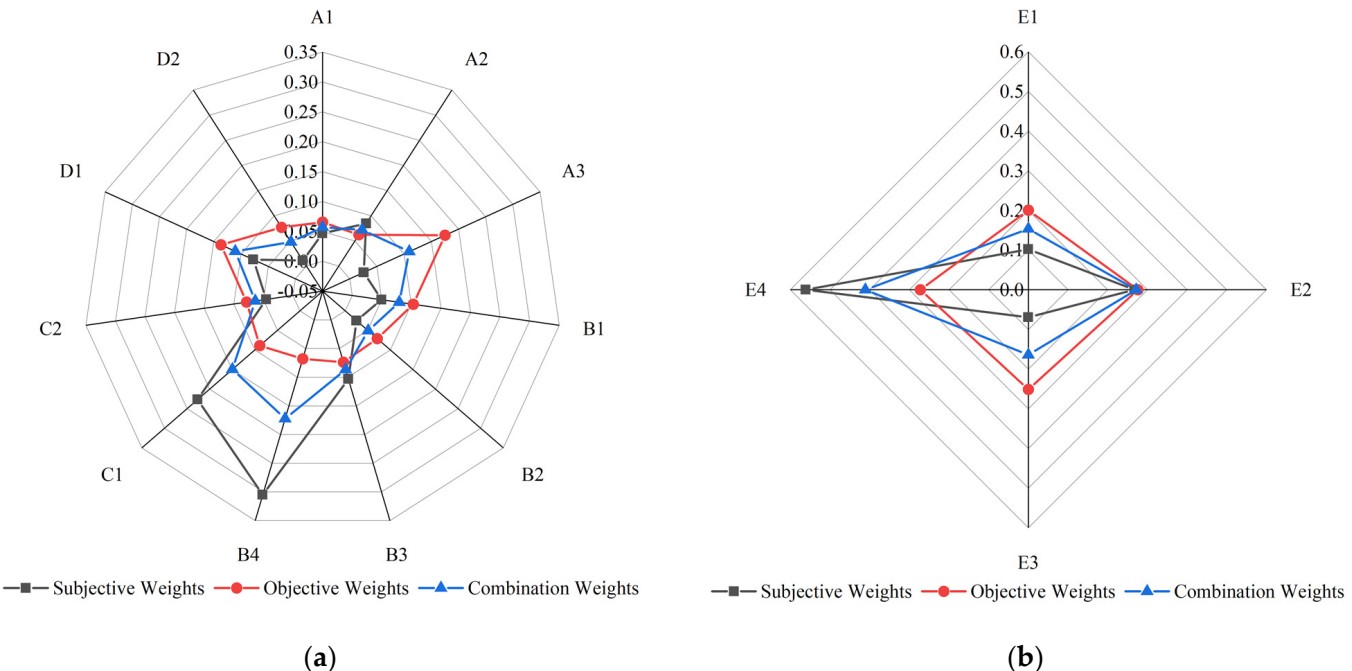

**Figure 4.** Weight proportion of each indicator. (**a**) Description of weight proportion of each indicator at target layer T1; (**b**) description of weight proportion of each indicator at target layer T2.

### 3.3. Individual Emergency Shelter

The relative superiority of a single emergency shelter is calculated and the evaluation scores are divided into five numerical intervals according to the natural interruption method, corresponding to five evaluation levels from low to high: unreasonable, comparatively unreasonable, generally reasonable, comparatively reasonable, and reasonable, and the evaluation results of the spatial distribution of each emergency shelter are obtained as shown in Figure 5.

From the scale of individual emergency shelters, there are nine emergency shelters in Shanghai with unreasonable spatial distribution, which need to be banned or improved; nineteen shelters are comparatively unreasonable and need to be adjusted; thirty-five shelters are generally reasonable; twenty-six shelters are comparatively reasonable; and two shelters are reasonable.

Combined with Table 4, it can be seen that most of the nine unreasonable places are due to the extreme unreasonableness of a certain evaluation indicator of the shelter, and the poor performance of several indicators leads to affect the overall spatial distribution evaluation results of the shelter. For example, for Tongchuan School, Nanpu Square Park, and Jinshan Stadium, the number of flammable and explosive places within a kilometer of the locations is as high as fourteen, ten, and seven, respectively, the highest of all emergency shelters in the study area, and the shelters have great potential safety disasters. The service potential crowdedness values of Tongchuan School and Hongkou Experimental School are as high as 7.7 and 10.8, respectively, the service potential crowdedness values are higher than the service limit at the time of shelter design, and the shelter suitability was not good.

Among the nine unreasonable shelters, except Nanpu Square Park, the service overlap area ratio is 0, and the rest are close to 100%. The spatial services of the shelters are highly redundant.

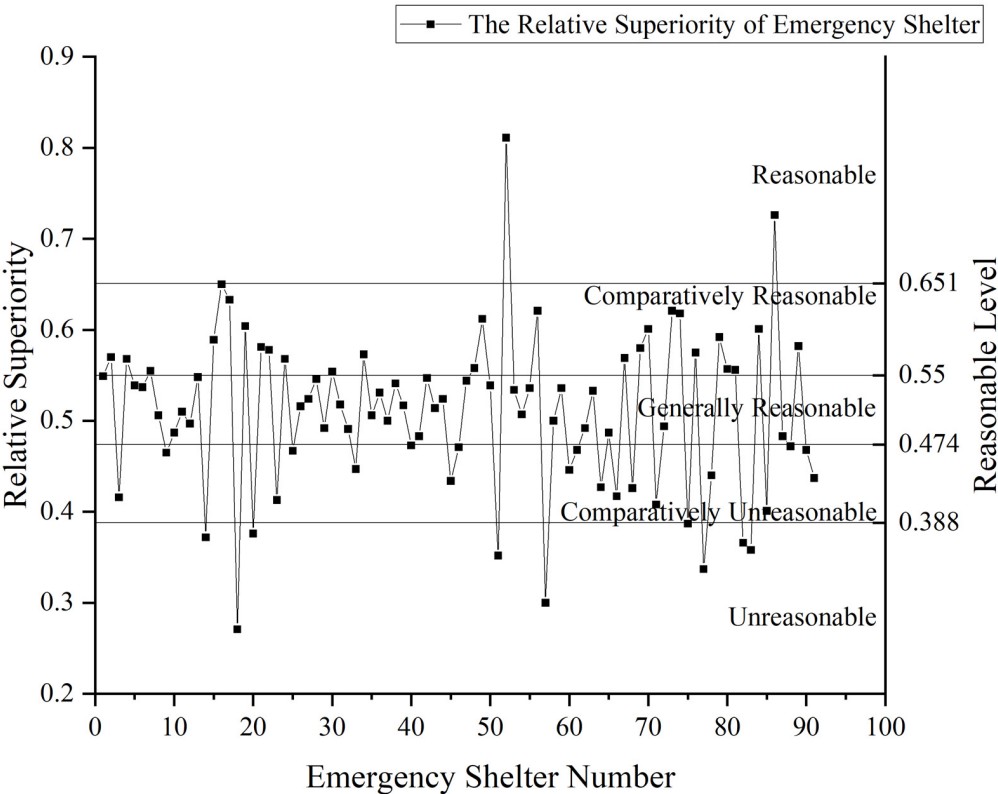

**Figure 5.** Evaluation results of the spatial distribution of individual emergency shelters.

*3.4. Regional Emergency Shelter Groups*

In the same way, the results of the spatial distribution evaluation under the effect of regional grouping of each emergency shelter are calculated as shown in Figure 6. From the scale of the regional grouping of emergency shelters, there is one region with unreasonable spatial distribution, which is Pudong New District, and its relative superiority value is much lower than other regions. Combined with Table 5, it can be seen that there is only one emergency shelter available in the region, and the number of emergency shelters is seriously insufficient and the service population gap is much larger than other regions, so it is urgent to plan new shelters to relieve the service pressure. There are three comparatively unreasonable regions, six generally reasonable regions, three comparatively reasonable regions, and three reasonable regions.

The spatial distribution of the evaluation results of emergency shelters is shown in Figure 7. Combined with Figure 7, it can be seen that, after the regional grouping of emergency shelters, their reasonableness evaluation scores show a high–low–medium distribution from the downtown area of Shanghai to the outside.

Several regions close to the downtown area of Shanghai (Xuhui, Changning, Jing'an, Hongkou, Putuo, Huangpu) are highly populated, the number and distribution of emergency shelters are denser, and the area of the region is smaller compared to the regions outside the downtown area, so the service area ratio, service population ratio, and service population gap indicators of the grouped emergency shelters in the region have more excellent performance and their reasonableness evaluation scores are higher.

The overall population density of the regions around the downtown area of Shanghai (Pudong, Minxing, Songjiang, Baoshan) is not as high as that of the downtown area, but there are some areas with high population density. However, the number and density of emergency shelters are much lower than in the downtown area, so this leads to a low evaluation score of the spatial distribution reasonableness. Especially in Pudong New District, the regional area is much larger than other regions. From Figure 3, we can see that Pudong New District still has a high population density distributed around the Huangpu River, but there is only one emergency shelter available in the whole region, so the evaluation score after the grouping of shelters in this region is the lowest in the city. The population density in the marginal areas of Shanghai (Jiading, Qingpu, Jinshan, Fengxian, and Chongming) is low, and most of the emergency shelters are established in densely populated areas. Therefore, the spatial distribution of emergency shelters under the effect of regional groups is comparatively reasonable.

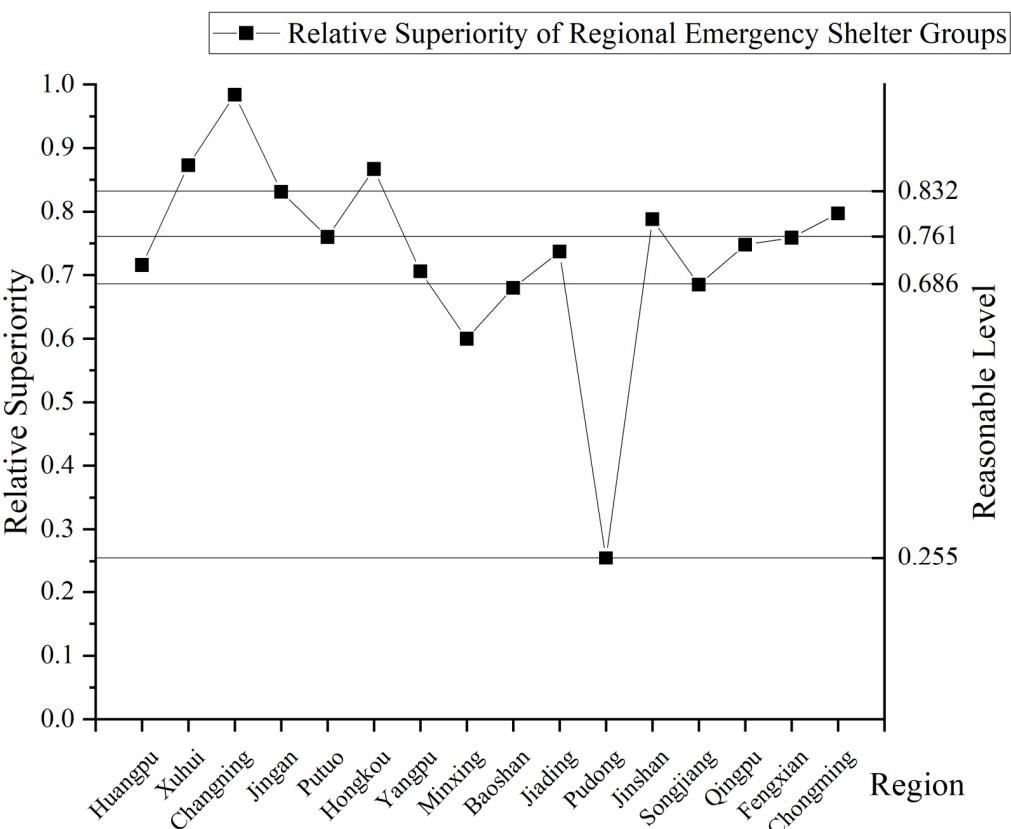

**Figure 6.** Evaluation results of regional groups of emergency shelters.

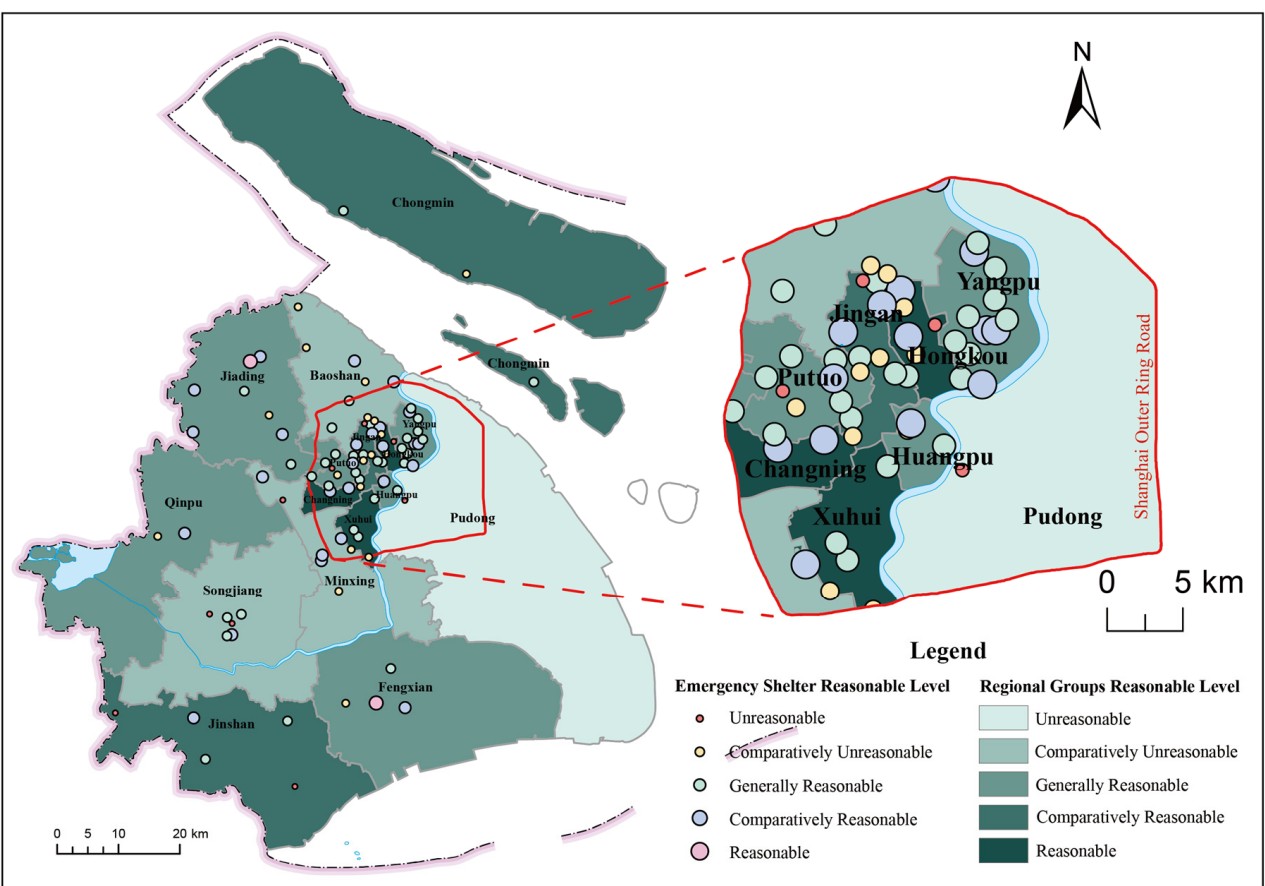

**Figure 7.** Spatial distribution of evaluation results of emergency shelters.

## 4. Discussion

### 4.1. Effectiveness of the Proposed Method

In the aspect of evaluation indicators, compared with previous studies [16–22], the method proposed in this paper expands on the selection of evaluation indicators for the spatial distribution reasonability of shelters in terms of evaluation scale and dimensions of evaluation indicators. In terms of evaluation scales, in addition to the evaluation of individual emergency shelters, the fairness of the spatial distribution of emergency shelters in different regions is also evaluated from the scale of regional groups of emergency shelters. In the evaluation dimension, the indicators under the traditional criteria of effectiveness, accessibility, and safety are improved, and the suitability and fairness criteria are added to reflect the service efficiency and fairness of emergency shelters under the current spatial distribution. Taking Shanghai as the study case for evaluation and analysis, the results show that most of the emergency shelters evaluated as unreasonable have problems in terms of effectiveness, suitability, safety, etc., which is mainly reflected in the high service overlap area of shelters, high potential crowdedness of services, the number of flammable and explosive places within a kilometer, etc., and the accessibility indicators of residents also reflect well the accessibility differences between different shelters. This indicates that the increased evaluation indicators can effectively reflect the existence of problems in the spatial distribution of emergency shelters. In terms of evaluation scale, the increase in the scale of regional groups of emergency shelters can effectively compensate for the problem that the evaluation of a single emergency shelter can hardly reflect the overall emergency shelter service capacity of the region, help to discover the differences in emergency shelter service capacity among different regions, and provide some reference for the optimization of the spatial distribution of emergency shelters.

In terms of evaluation methods, there are problems, such as single weights in previous studies and ignoring the influence of fuzzy incompatibility between different attribute indicators on the comprehensive evaluation results in multi-indicator decision-making [25–31]. To address the problems, this paper optimizes the weight allocation of the evaluation indicators by the AHP method, CRITIC method, and the optimal weight coefficient solution method based on the sum of squares of the maximum deviation and constructs a multi-indicator evaluation model for the spatial distribution reasonability of emergency shelters based on fuzzy theory. In order to prove the validity of the methods used in this paper, the AHP method and TOPSIS method, which are commonly used at present, were selected to compare with the methods used in this paper under the same evaluation indicator system, and the statistical results of the evaluation scores of the three methods under the two evaluation scales were compared in Tables 6 and 7, respectively. As can be seen from the table, the coefficient of variation of the evaluation scores of emergency shelters calculated by the evaluation method used in this paper is significantly greater than that of the AHP and TOPSIS methods. This indicates that the evaluation scores calculated by the method used in this paper have a higher degree of dispersion, and the evaluation of the spatial distribution reasonability of the emergency shelter is more discriminative. This confirms that the method used in this paper can well solve the influence of the fuzzy incompatibility between indicators on the comprehensive evaluation results and improve the reliability of the evaluation results.

**Table 6.** Statistics of evaluation results of three methods under target layer T1.

| Method | Minimum | Maximum | Average | Standard Deviation | Variable Coefficient |
|---|---|---|---|---|---|
| This paper | 0.2713 | 0.8106 | 0.5101 | 0.0868 | 17.01% |
| AHP | 0.2643 | 0.7502 | 0.4600 | 0.0063 | 14.41% |
| TOPSIS | 0.4780 | 0.7150 | 0.5517 | 0.0384 | 6.97% |

**Table 7.** Statistics of evaluation results of three methods under target layer T2.

| Method | Minimum | Maximum | Average | Standard Deviation | Variable Coefficient |
|---|---|---|---|---|---|
| This paper | 0.2537 | 0.9845 | 0.7366 | 0.1576 | 21.41% |
| AHP | 0.2702 | 0.9127 | 0.6884 | 0.1376 | 19.99% |
| TOPSIS | 0.3690 | 0.8510 | 0.5867 | 0.1218 | 20.76% |

The above comparative discussion results further show that the multiple evaluation indicator system established in this paper can effectively reflect the defects in the spatial distribution of the existing emergency shelters in Shanghai, and the constructed multi-indicator evaluation model can effectively evaluate the spatial distribution reasonableness of the existing emergency shelters in Shanghai.

### 4.2. Limitations of the Method

The method proposed in this paper has improved compared with previous research in terms of the evaluation system and evaluation method. However, due to the difficulty of obtaining data sources, there are still some limitations in applicability.

In the simulation of the geospatial distribution of urban population, this paper uses the residential neighborhood scale as the demographic unit. Because of the day and night population movement, the applicability of the method will be different. The results would be even better if we could capture the geospatial distribution of urban population in real time. In addition, the evaluation indicators selected in this method consider the common situation of most disasters. For some special disasters, it is necessary to consider separate evaluation indicators, such as the drainage capacity of flood disaster shelters.

## 5. Conclusions

This study proposes a method to evaluate the reasonableness of the spatial distribution of urban emergency shelters. In response to the problems in previous studies, this paper enriches the evaluation scale and evaluation indicator dimension so that it can reflect more comprehensively the deficiencies in urban emergency shelters in terms of spatial distribution. The constructed multi-indicator evaluation model optimizes the rationality of weight allocation, overcomes the influence of ambiguity and uncertainty among different attribute indicators on the evaluation results, and makes the obtained evaluation results more scientific and reasonable.

From the evaluation results, among the ninety-one existing emergency shelters in Shanghai, there are nine shelters with unreasonable spatial distribution, nineteen shelters with comparatively unreasonable distribution, thirty-five shelters with generally reasonable distribution, twenty-six shelters with comparatively reasonable distribution, and two shelters with reasonable distribution. Analyzed from the scale of the regional grouping of emergency shelters, there is one district with unreasonable spatial distribution, which is Pudong New District, and its relative superiority value is much lower than that of other districts; three districts with comparatively unreasonable spatial distribution; six districts with generally reasonable spatial distribution; three districts with comparatively reasonable spatial distribution; and three districts with reasonable spatial distribution. Moreover, the reasonableness values of each district show a high–low–medium distribution from downtown Shanghai outward.

**Author Contributions:** Conceptualization, M.G., C.D. and X.W.; methodology, X.W.; data curation and visualization, F.X. and X.W.; writing—original draft preparation, X.W.; writing—review and editing, X.W., M.G., J.W., Y.F. and G.L. All authors have read and agreed to the published version of the manuscript.

**Funding:** This work was supported by the Youth Fund of Shenzhen Polytechnic (6022310015K), Scientific Research Launch project of Shenzhen Polytechnic (6022312051K), Guangming Laboratory Open project (GML-KF-22-21), National Key Research and Development Program (2019YFB2102703), Natural Science Foundation of Guangdong Province (Grant 2019A1515011267), Project of Educational Commission of Guangdong Province (Grant 2021ZDZX1090), Shenzhen Basic Research Project (Grant JCYJ20190809113617119), Shenzhen Polytechnic Project (Grant 6021310017K, 6022310032K), and Lihu Elite Training Project (Grant LHRC20220409).

**Data Availability Statement:** All open datasets for this study are available from the corresponding author upon reasonable request.

**Acknowledgments:** The authors acknowledge all the data sources mentioned in the article for providing the open-source big data and are thankful for all the helpful comments provided by the reviewers and editors.

**Conflicts of Interest:** The authors declare no conflict of interest.

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
