# Peer review of "A Multi-Indicator Evaluation Method for Spatial Distribution of Urban Emergency Shelters"

_remotesensing, doi:10.3390/rs14184649_

Round 1

Reviewer 1 Report (Previous Reviewer 4)

I have no more comments on this manuscript.

Author Response

Reviewer 2 Report (Previous Reviewer 1)

This manuscript submitted has been improved with the addition of new analysis and the change of English expression. The author has answered all questions raised by the reviewers and made a significant attempt to improve the manuscript. I just have one addition comment, requiring a minor revision: it should be “km” instead of “Km” in the scale.

Some references maybe useful for your manuscript revisions.

1)     Decrease in the residents' accessibility of summer cooling services due to green space loss in Chinese cities

2)     Effects of urban street trees on human thermal comfort and physiological indices: a case study in Changchun city, China

Author Response

This manuscript is a resubmission of an earlier submission. The following is a list of the peer review reports and author responses from that submission.

Round 1

Reviewer 1 Report

This paper used the indicator system method used to reflect the deficiencies in the spatial distribution of urban emergency shelters, however a few missing issues that would enhance the study.

Issue1: Please add the discussions on the results from this study and other researches.

Issue2: Paragraph 2 in the introduction: The language is cumbersome, and the application of gis to geospatial analysis does not need much description.

Issue3: It is recommended that the introduction provide a summary of past research rather than a brief commentary, particularly, lines 63-65. Can the single and regional spatial distribution evaluation scales mentioned in the article solve the problem that there is no unified index system for evaluating the spatial distribution of urban emergency shelters?

Issue4: The graphic presentations fall short in some ways, including the following: not being readable due to tiny font sizes; lacking of scale bar in Chinese maps (cf. Fig 1); the color of class â…¢ is indistinguishable from the population density (cf. Fig 2); not be accessible to the colorblind due to using red and green together (cf. Fig 3).

Issue5: I suggest a revision in the Results and Analysis. The analysis does not look deep enough.

Issue6: Discussion should place the major findings into perspective of current literature in the subject and not reiterate the results. The content of the discussion is not sufficient.

Line 56: What is spatial complementarity for a single-scale study, and what is it aimed at? Do multi-scale studies address these questions?

Line 168-171: Is the kernel density function applied by the authors a good fit for the attenuation effect of people arriving at emergency shelters and is it scientifically sound?

Lines 442: needs rephrasing

Line444: Why used the natural interruption method? It is debatable whether this method is really meaningful for the division of sanctuary plausibility. Because this method is based on the results of the reasonableness classification of shelters in Shanghai that the authors have already calculated, such a classification is relative, and other cities do not apply the values of these rankings.

Lines 447: What’s “results of the spatial distribution”, I don't think Figure 4 is a spatial distribution map.

Reviewer 2 Report

The authors presented an interesting concept for studying the Spatial Distribution of Urban Emergency Shelters. The problem is socially important. For this purpose, they chose a large group of indicators to evaluate their effectiveness, accessibility, and safety. Of note are the very good maps. The results have been well demonstrated.  

The article is very interesting but needs clarification as not everything is understood by the reader. 

1. The authors mention several times in the text about previous studies without providing literature, the reviewer does not understand whether these were other studies by the same authors or others? 

2. The article uses many indicators, in addition to describing the indicators, what is missing from the literature review are papers on these indicators, discussions, and examples of their application (not necessarily to the study of shelters). 

3. There is too little information in the text about the ArcGIS spatial analysis functions used, and they were treated too generally. Were the analyses performed exclusively in ArcGIS? In what software were the indicators calculated?

4. Did the authors assume that the population would travel by car to the shelters. Why didn't they consider local pedestrian and bicycle routes? 

5. How was the value of the variables determined? For example, in formula (1).

6. Each coefficient should be linked to the scientific literature, as they are not authored by the authors of this publication  

7. I believe that the introduction lacks a discussion of the multivariate methods of assessing the spatial distribution of other point elements in the city (with or without weighting). This is the weakest part of the article. 

8. I propose that the data sources be collected in a table, which will be given the format of the data, the scale, the source of access with the date of downloading

9. A discussion of the data would be useful. Their advantages and disadvantages. 

Reviewer 3 Report

1. Introduction

The authors need to conduct a more extensive review of the relevant articles involved, especially for relevant studies on this topic in the target journal. If there is no literature that can be cited, authors need to consider the suitability of the journal. In addition, authors need to highlight the wide-ranging merits of the work.

2. Principles and Methods

The authors want to underline the novelty of the method and further simplification is recommended for irrelevant notations and descriptions. It is possible to use cited references. It is not necessary to give every notation and procedure.

It is recommended to add an overall technical diagram. The numbering in the text is easily confused with the headings.

3. The third section, Data Sources and Analysis, and the fourth section, Results and Analysis, need to be combined or separated clearly.

4. The discussion section is rather over-simplified. The method is the point of innovation in the meaning of the way this paper is presented. If so, comparative studies using different methods are needed.

Reviewer 4 Report

This paper evaluates the spatial distribution condition of the existing 91 emergency shelter in Shanghai, China. However, the organization and presentation of this paper need a substantial improvement.

1.     Fig. 1, 2, 6: Try to organize them more clear.

2.     Line 359-427: This paper just provided the data source rather than how to generate the indicators. Moreover, validations of the obtained data for the following analysis were also essential. In addition, Remote Sensing is a journal in the field of remote sensing. So how remotely-sensed data can help you carry out this work? For instance, obtain indicators, spatial analysis, or validation the results?

3.     Line 383: 3.3 Data Analysis, This section is about how to analyze the data and generate the results (i.e., methods you used) rather than the exhibition of the results of your data. For example, how to weight the indicators used in this paper (4.1 Indicator Weighting)? how to define and classify the evaluation results of the regional groups of emergency shelters (Line 479: Fig. 5)? How to choose the method for assessment (why you choose the AHP method, as I known, this is a subjective evaluation method.)

Round 2

Reviewer 1 Report

The structure of this paper is not reasonable. The study area and dataset should be put into the part of method. Besides, there are serious problems in the part of Methods. Just as the limitation the authors mentioned  in discussion.

Reviewer 4 Report

This article still needs great improvement according to the prior first-version comments.
For example: comment 1 “ Fig. 1, 2, 6, 7: Try to organize them more clear.”: Authors should explain why they just reorganized the Fig.1 and Fig. 2 rather than all the unclear figures.
Comment 2 “Line 359-427: This paper just provided the data source rather than how to generate the indicators. Moreover, validations of the obtained data for the following analysis were also essential. In addition, Remote Sensing is a journal in the field of remote sensing. So how remotely-sensed data can help you carry out this work? For instance, obtain indicators, spatial analysis, or validation the results?”: Please consider all the comments and recommendations rather than just exhibit the difficulties you encountered. If you think some comments are confusing, please clarify why they are confusing.
Comment 3 “Line 383: 3.3 Data Analysis, This section is about how to analyze the data and generate the results (i.e., methods you used) rather than the exhibition of the results of your data. For example, how to weight the indicators used in this paper (4.1 Indicator Weighting)? how to define and classify the evaluation results of the regional groups of emergency shelters (Line 479: Fig. 5)? How to choose the method for assessment (why you choose the AHP method, as I known, this is a subjective evaluation method.)”:

Please try to confront all these comments and make some clarifications rather than delete the “Data Analysis” section. Because I think the “Data Analysis” section is an important part of your paper.